# Tailoring electron beams with high-frequency self-assembled magnetic charged particle micro optics

R. Huber[1,2,7], F. Kern[3,7], D. D. Karnaushenko [1,4], E. Eisner[1], P. Lepucki[3], A. Thampi[3], A. Mirhajivarzaneh[1], C. Becker [1,4], T. Kang [1], S. Baunack [1], B. Büchner[3,5], D. Karnaushenko [1,4✉], O. G. Schmidt [1,2,4,6✉] & A. Lubk [3,5✉]

Tunable electromagnets and corresponding devices, such as magnetic lenses or stigmators, are the backbone of high-energy charged particle optical instruments, such as electron microscopes, because they provide higher optical power, stability, and lower aberrations compared to their electric counterparts. However, electromagnets are typically macroscopic (super-)conducting coils, which cannot generate swiftly changing magnetic fields, require active cooling, and are structurally bulky, making them unsuitable for fast beam manipulation, multibeam instruments, and miniaturized applications. Here, we present an on-chip micro-sized magnetic charged particle optics realized via a self-assembling micro-origami process. These micro-electromagnets can generate alternating magnetic fields of about ±100 mT up to a hundred MHz, supplying sufficiently large optical power for a large number of charged particle optics applications. That particular includes fast spatiotemporal electron beam modulation such as electron beam deflection, focusing, and wave front shaping as required for stroboscopic imaging.

--------

[1] Institute for Integrative Nanosciences, Leibniz IFW Dresden, 01069 Dresden, Germany. [2] Material Systems for Nanoelectronics, Chemnitz University of Technology, 09107 Chemnitz, Germany. [3] Institute for Solid State Research, Leibniz IFW Dresden, Helmholtzstraße 20, 01069 Dresden, Germany. [4] Center for Materials, Architectures and Integration of Nanomembranes (MAIN), Chemnitz University of Technology, 09126 Chemnitz, Germany. [5] Institute for Solid State and Materials Physics, TU Dresden, Dresden, Germany. [6] Nanophysics, Faculty of Physics, TU Dresden, 01062 Dresden, Germany. [7] These authors contributed equally: R. Huber, F. Kern. ✉email: d.karnaushenko@ifw-dresden.de; oliver.schmidt@main.tu-chemnitz.de; a.lubk@ifw-dresden.de

Charge particle optical devices for electron beam steering and shaping, such as lenses, stigmators, and deflectors, are essential components in electron microscopes, lithography instruments, and colliders, which have been elaborated in recent decades[1–5]. Previous miniaturization efforts in charged particle optics (CPO) focused mainly on electric field generating microstructures implemented in the form of accelerators[6], lenses[7,8] their assemblies[9] as well as phase plates[10–13]. Miniaturized magnetic charged particle optics (MCPOs) have been realized using permanent magnets[14–16] and planar lithographic techniques[17]. Relying on miniaturized CPO devices novel electron microscopes[9], accelerators[6,14], electron beam lithography instruments, and free electron lasers (FEL)[15] were realized. However, due to the large fields required at high particle velocities, charging problems, and limited dielectric strength of the isolating materials and the high vacuum, the electrostatic CPO devices mostly operate in the low particle energy regime (e.g., <10 kV for electrons) or as low optical power devices (e.g., phase plates). Macroscopic MCPO devices do not allow fast beam modulation due to the use of permanent magnets, large inductances, inherent losses, and Ohmic heating. These persisting drawbacks call for the development of miniaturized MCPO devices capable of modulating charged particles at high velocities and high frequencies with high optical power, e.g., in electron microscopes.

CPO devices that generate transverse magnetic fields have high optical power because the Lorentz force is proportional to the particle velocity. Thus, transverse MCPOs are ideal candidates for realizing miniaturized CPO devices that can operate at large particle energies[14,15,17]. However, the manufacturing of transverse microsized MCPOs (μMCPOs) faces a number of technological challenges associated with the realization of small-scale electromagnetic coils capable of generating sufficiently strong magnetic fields in a small, confined volume of a few 1000 μm$^3$. Recently developed self-assembling micro-origami technologies[18–20] offer means to realize microscale cylindrical architectures with integrated electronic functionalities such as wafer-scale integrated microscale capacitors[21], sensors[18,22,23], electromagnetic coils[24–26], and microsystems[27]. Self-assembled micro-coils have been shown suitable for generating strong magnetic field in a small volume, as required in nuclear magnetic and electron spin resonance (NMR and ESR) spectroscopies[25,26]. The self-assembled micro-coils[24] have already been proven to operate up to the GHz regime, benefiting from vanishingly small inductances and reduced parasitic effects compared to their larger counterparts[26].

In this work, we use wafer-scale self-assembled micro-coils as magnetic field sources that are driven by an external current source via on-chip integrated electric feedings allowing us to control strength, polarity, and frequency of the generated magnetic field. The micro-coils are equipped with ultrasoft-magnetic micro-wire cores[28,29] that guide and concentrate the magnetic field towards the electron beam locus (see Fig. 1a). The resulting μMCPOs have high optical power and can operate at high frequencies without the need for water cooling, heavy mechanical supports, and large vacuum infrastructure that conventional large MCPO elements require. We developed three μMCPO variants, namely an electron vortex phase plate (generating an electron beam with orbital angular momentum (OAM)), a uniaxial dipole assembly of magnets, and a quadrupolar element with four biaxial magnets. These μMCPOs modulate the electron beam in the spatiotemporal domain, allowing for fast deflection, focusing, and shaping of electron beams even in tiny spaces like those in transmission electron microscope aperture planes (see Fig. 1b). Among other applications, the high-speed electron beam modulation (Fig. 1c) opens up intriguing opportunities to perform stroboscopic experiments with a narrow pulse width while maintaining high intensity and repetition rate.

## Results

**Fabrication and yield of self-assembled coils.** The fabrication of self-assembled micro-coils in μMCPO devices requires the synthesis and subsequent patterning of functional polymeric films[18–20,24], namely the sacrificial layer (SL), the hydrogel (HG), and the polyimide (PI) layers on top of a pretreated 150 mm (6-inch) wafer. Then a Ti–Cu–Ti layer (thickness ca. 200 nm) is patterned with a lift-off technique to form the electrical conducting structure of the coils as shown in Fig. 2a (more details are provided in "Methods" and in Supplementary Note 1, see also Supplementary Fig. 1). Then the Bosch Etching Process (BP) is applied to the backside of the wafer to open the aperture that allows the electron beam to pass through the wafer itself. Afterwards, the SL is selectively etched away to release the above-lying stack and simultaneously swell the HG layer. Being reinforced by the PI layers, the PI/HG/PI structure generates differential stress that is released when the material stack reshapes into a "Swiss-roll" architecture, with one edge of the structure remaining fixed at the substrate surface (Fig. 2b). The self-assembly direction is guided by introduced stripes of PI perpendicular to the rolling direction (Fig. 2b) as previously reported[25,30]. Figure 2c shows that the entire microfabrication process yields up to ~600 samples simultaneously on a 6-inch wafer. On such a wafer, 1214 planar coils were electrically tested, with two having damaged loops. After the self-assembly process was finished, 28 coils were not electrically functional and 45 structures did not self-assemble properly, achieving in total a 91.4% yield. The distributions of resistance and inductance before and after the rolling process are shown in Supplementary Fig. 2. More information about the manufacturing yield can be found in Supplementary Note 2.

Each μMCPO element has several micro-coils, however, only those adjacent to the aperture opening are of interest. These coils are manufactured from a rolled copper stripe (50 μm × 2.6 mm and 280 nm thick) to maximize the transverse magnetic field flux density (Fig. 2d) when excited with $I_c$, i.e., the electric current through the coil. Indeed, finite element method (FEM) simulations (Fig. 2e) reveal that micro-coils with a 140 μm diameter at 100 mA current can generate magnetic flux densities of up to 3 mT next to the coil edge. In terms of current intensity, it is critical to avoid micro-coil failure due to a heat-driven breakdown; see the current breakdown graph and the effects of overheating the micro-coil in Supplementary Fig. 2b, c, respectively. Figure 2f shows a mean breakdown current of 93.0 mA ± 12.7 mA among all the properly assembled coils from a single 6-inch wafer. This current is sufficient for generating a magnetic field capable of re-magnetizing the soft magnetic micro-pole in the electromagnet (Fig. 1b) and we used only those coils for further processing that withstand the 100 mA limit. The micro-pole tips are structured via focused ion beam milling (FIB) (Fig. 2g, inset), in which a micro-wire, previously fixed to the coils with polymeric adhesive, is shaped into the desired geometry of the pole tip (see Supplementary Fig. 3). We use special glass-coated micro-wires composed of amorphous CoFeSiB, where the manufacturing process (wire pulling inside the glass envelope) induces a small circular magnetoelastic anisotropy, leading to a magnetic vortex state, which may be easily magnetized along the wire[28]. Indeed, Fig. 2g shows that the magnetization of the ultrasoft ferromagnetic micro-wire spans from saturation to saturation almost linearly without large hysteresis effects while being bidirectionally swept in a small homogeneous external field ranging from −0.2 to +0.2 mT (see "Methods"). Note, furthermore, that the glass

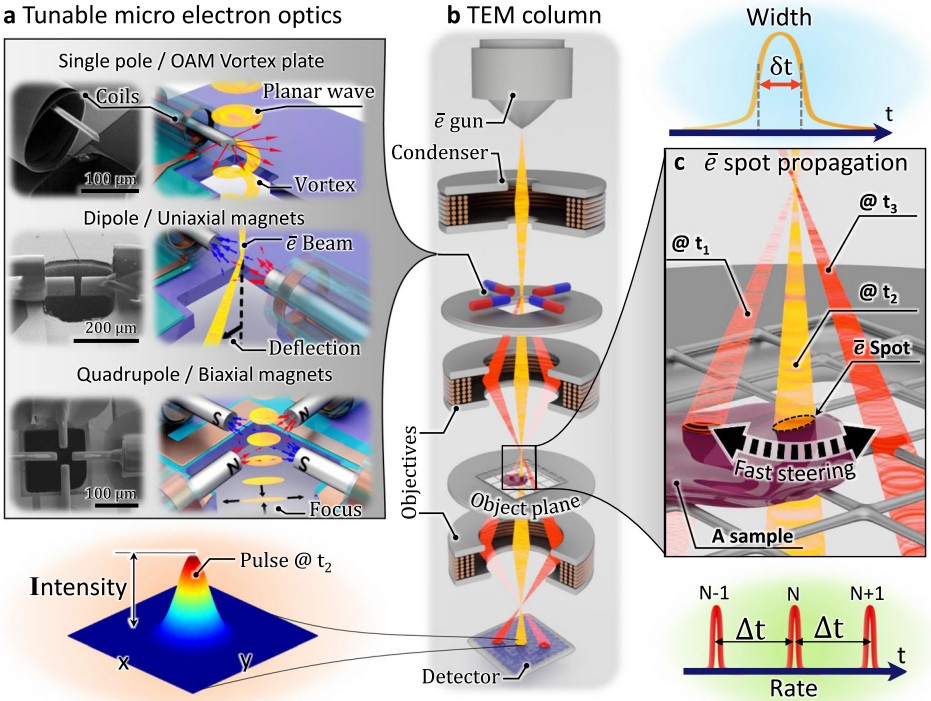

**Fig. 1 Tunable microsized magnetic charged particle optics for high energy electron beam systems such as transmission electron microscopes.**
**a** Micro-electromagnets designed in three variants: a single pole that can operate as an electron vortex phase plate; a dipole that is built from two collinear micro-coils (two uniaxial magnets); a quadrupole assembly (four biaxial magnets) that can be configured to either generate a strong flux gradient needed to perform focusing functions or to generate an in-plane vector field whose orientation can be changed dynamically. **b** Simplified sketch of a TEM column demonstrating the principle of electron beam modulation with microscale electromagnets inserted in an aperture plane of the microscope. **c** Micro electro magnets provide a unique possibility to steer the beam with high rate providing means for high speed stroboscopy with a narrow and intense electron beam pulse probing the sample.

envelope additionally serves as electrical insulator preventing any short-cut of the coils by the soft magnet filling.

**Monopole and dipole device.** After wafer dicing a micro-wire is inserted and FIB milled in a preselected micro-coil that could withstand 100 mA current, which is the limit for the Protochips Fusion™ TEM specimen holder (Fig. 3a). The resulting single pole element with a tapered micro-pole (Fig. 3a, bottom inset) is installed in the TEM specimen holder, allowing in situ electrical contacting of the chips within a FEI Titan³ 80–300 Transmission Electron Microscope (TEM). We demonstrate that a tunable micro-coil device can effectively modulate an electron beam accelerated up to 300 kV under high vacuum conditions, and no signs of degradation are revealed after several days of operation.

The performance of the single pole vortex µMCPO was evaluated by scanning a focused electron beam over the micro-pole tip region and measuring the deflection in the far field. In this so-called differential phase contrast (DPC) setup the deflection was determined from the shift of the center of mass of the diffraction disk in the far field of the device (see Supplementary Note 3). Figure 3b depicts the 2D maps of the measured magnetic fields where the amplitude of one projected **B**-field component for $x$ and $y$ directions are shown at different $I_c$ values and the corresponding **B**-field vectors.

The magnetic field profile of a single pole µMCPO is suitable to generate a (half) electron vortex beam[11] of a very high orbital angular momentum (OAM > $10^4$ $\hbar$), which exceeds any achieved OAM reported for electron beam vortex generating devices by roughly one order of magnitude[12,31–33]. Figure 3c shows the evaluation of the DPC in terms of OAM by computing

$\varphi = \oint d\varphi \approx 10^4 \cdot 2\pi$ along a "closed" contour connecting the left and the right side of the micro-pole (see Supplementary Note 3). The OAM of the beam can be continuously modulated by changing the excitation current of the micro-coil as shown in Fig. 3d, where $I_c$ ranges from +45 to −15 mA. One readily observes the typical doughnut-like shape of the vortex beam with zero intensity in the center caused by the destructive interference. However, the focused beam is not exhibiting a full circle, which we ascribe to the finite thickness of the magnetic wire, electrical charging[34] or magnetic fields emerging along the tapered shape of the tip (see Supplementary Note 4). These effects finally render the Dirac string approximation, and hence the vortex beam, incomplete, which may be further improved by tailoring the shape of the micro-wire tip by FIB shaping. We finally note that the OAM imparted on the electron leads to an orbital spin Hall effect when deflected in an electric field[35] and a Zeeman effect when traversing magnetic fields parallel to the OAM[36]. These and other effects scale with the OAM of the electron vortex beam, thus making high intensity and high OAM electron vortex beams a unique experimental probe in elaborate electron optical systems such as electron microscopes.

Aligning two micro-electromagnets in one device yields a dipole (Fig. 3f, inset), which can be employed as a charged particle beam deflector. This assembly is installed in the same TEM specimen holder for characterization purposes in an analogous manner to the single pole. Again, DPC measurements allow to characterize the whole magnetic field in the device and to calculate the deflection angle $\alpha$ of the beam from the camera length $L$ and the displacement $d$ of the beam on the detector (see Fig. 3e, further details in the Supplementary Note 3). The relationship of the deflection angle $\alpha = eB_t/m_e v$ to the projected

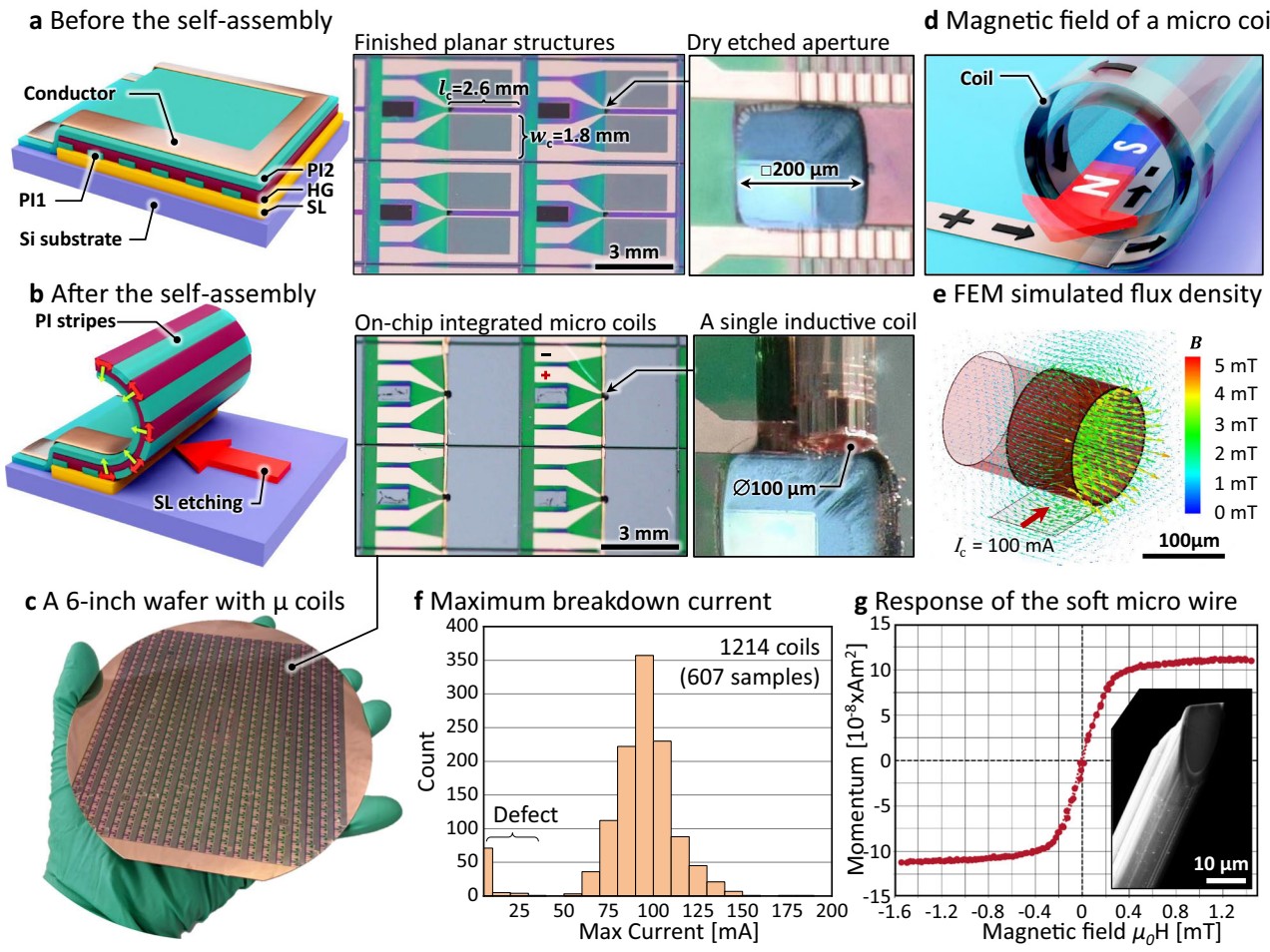

**Fig. 2 Self-assembled micro-coils (exemplified for coil pair sample design). a** Sketch of the microfabricated layer stack and structures before the self-assembly process with a micrograph of four adjacent structures on the wafer. The planar structure has a width of $w_c = 1.8$ mm and is as long as the conductor stripe (length $l_c = 2.6$ mm) with a width of 50 μm, which later on forms the micro-coil. The structures contain openings forming an aperture for the electron beam. **b** Sketch and micrograph of the self-assembled micro-coils which are produced by etching of the sacrificial layer (SL) and swelling the hydrogel layer (HG). The conducting metal layer is placed on top of a polyimide layer (PI) **c** 6-inch wafer with 1069 out of 1214 functional integrated micro-coils. **d** Generation of magnetic field with a micro-coil when current passes through the conductor. **e** FEM simulation of the magnetic flux density generated by a coil driven by $I_c = 100$ mA current shows that the coil can provide high fields of about 3–4 mT. **f** Distribution of the maximum breakdown current for the micro-coils fabricated on one wafer reveals a mean break down current of 93.0 mA. **g** The axial magnetic response of the soft ferromagnetic microwire (inset) with the transversal anisotropy reveals almost linear hysteresis free field dependence in the range from −0.2 to 0.2 mT external field.

field $B_t$ for an electron of relativistic mass $m_e$, velocity $v$, and elementary charge $e$ can be used to estimate the magnetic field generated by the μMCPO. Figure 3f shows that $\alpha$ ranges from −1.04 mrad at −100 mA to 0.64 mrad at 100 mA, which translates into an almost linearly tunable magnetic field ranging from −109 to 68 mT in the middle of the poles. The hysteresis of the deflections and hence the generated magnetic fields do not exhibit any measurable remanence in accordance with the ultrasoft magnetic properties of the circular anisotropic micro-wires reported before[28] and our measured magnetization characteristics (Fig. 2g). The bias of approximately 30 mA in Fig. 3f is presumably due to technological reasons pertaining, e.g., to inhomogeneities of the wire, FIB artefacts (not symmetric cut, Ga implantation in the wire), or slightly unmatched coil geometries.

**Quadrupole device**. Finally, by employing four micro-electromagnets arranged around an aperture opening, we created quadrupole μMCPOs (Fig. 4a), a key focusing element along with classical round lenses. Compared to round lenses, magnetic quadrupoles possess stronger optical power, thus they are more

useful at high particle energies and can additionally be used as stigmators and anisotropic focusing elements. Additionally, round orthomorphic focusing of a charged particle beam can be achieved by an assembly of three or more quadrupole lenses. A quadrupole μMCPO was installed in the TEM specimen holder like the previous samples and modulated with continuous current. Figure 4b shows the defocused beam profile in the far field. Without electric stimulation, the beam has a round shape, but when the micro-electromagnets are stimulated in the quadrupole configuration with $I_c = \pm50$ mA the beam undergoes a convergent action in a plane ±45° inclined to the orientation of the quadrupole and a divergent action in the perpendicular plane. This is congruent to the result of the DPC measurement in Fig. 4c. The focal length of such a thin quadrupole can be estimated as $f = m_e v / \left( e \int_{-\infty}^{\infty} C_{QP}(z) dz \right)$, where $C_{QP}$ denotes the $z$-dependent quadrupole strength containing fringing fields. For our particular system the measured focal length is 46 mm ± 8 mm at 300 kV accelerating voltage and 100 mA coil excitation (see Supplementary Note 5 for more details). Employing a rectangular field approximation of effective width 20 μm (i.e., the diameter of the poles) that corresponds to a quadrupole strength of

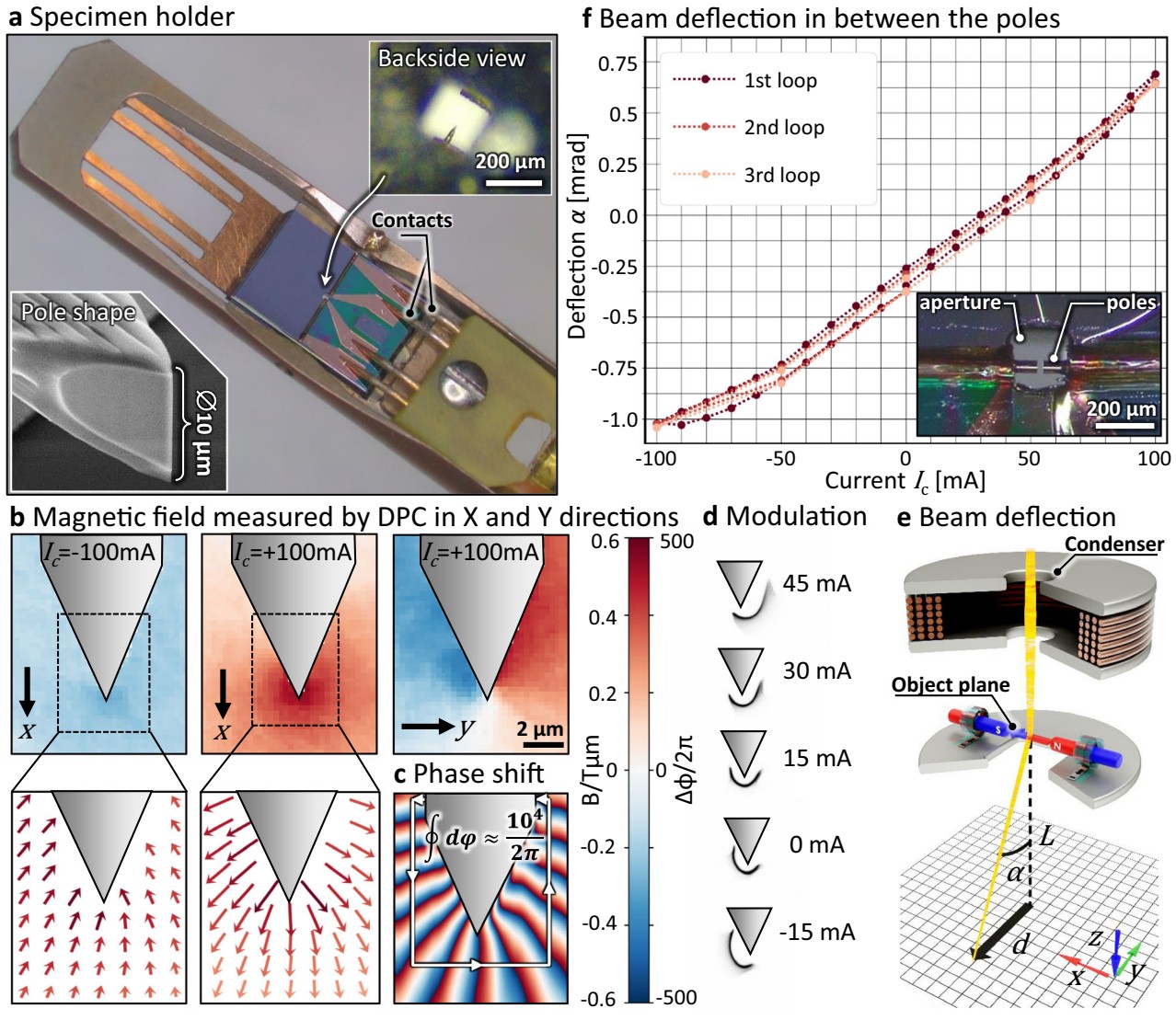

**Fig. 3 Electron optical characterization of µMCPO devices; electron vortex phase plate and dipole. a** Protochips Fusion™ TEM specimen holder with an inserted and electrically contacted chip carrying a complete set of self-assembled micro-coils and one micron-sized magnetic pole. The pole, made of a soft magnetic micro-wire, was tapered towards the tip (SEM overview in the bottom inset). The electron beam passes through the aperture in the Si chip (as shown in top inset). **b** Projected **B**-field components in x and y direction at different excitation currents $I_c$, and the corresponding **B**-field vectors, as determined by differential phase contrast (DPC) measurements. **c** Phase of the electron wave at 100 mA coil excitation as reconstructed from the DPC measurements. The integration path for computing the vortex orbital angular momentum (OAM) is indicated revealing a strong >$10^4$ OAM. **d** A series of electron (half) vortex beams in the far field (indicated defocus of −0.4 µm) at varying currents, demonstrating the functionality of the adaptive vortex phase plate allowing to modulate the electron beam and OAM on-the-fly. **e** Schematics inside of a TEM of a dipole; the micro-coils generate a magnetic field transversal to the e$^−$ beam; the deflection $\alpha$ can be derived from the camera length L and the displacement d of the center of mass of the beam measured on the detector in the far field. **f** DPC measurements of the deflection angle of the beam in between the poles of the µMCPO including variations at different sweeps exhibiting a nearly anhysteretic magnetization behavior.

$C_{QP} \approx 800$ T/m, which is in line with the quadrupole strength estimated from the DPC data and roughly one order of magnitude larger than previously reported microsized CPOs[17]. Accordingly, the actual achieved focal length is comparable in magnitude to conventional macroscopic focusing devices (both round lenses and quadrupoles) employed in electron microscopes, energy filters, or colliders. The imperfections (aberrations) in the system originate from the suboptimal alignment of the 4 poles (not forming a perfect quadrupole, see Supplementary Fig. 3f). They may be further reduced by improved fabrication and individually modulating the micro-coil strengths (currently

the quadrupole is excited symmetrically due to the limited number of connectors on the Protochips Fusion™ platform).

**High-frequency operation of µMCPOs.** One intriguing effect when miniaturizing electromagnets is the greatly enhanced frequency transfer due to the linear scaling of the corresponding inductances. In order to exploit the high-frequency transfer of the µMCPOs, we built a dedicated high-frequency biasing aperture holder (Fig. 4a), which allows to insert the µMCPOs in different aperture positions of the FEI Titan[3] TEM (see "Methods"). Moreover, by exciting opposite poles of the quadrupole µMCPO

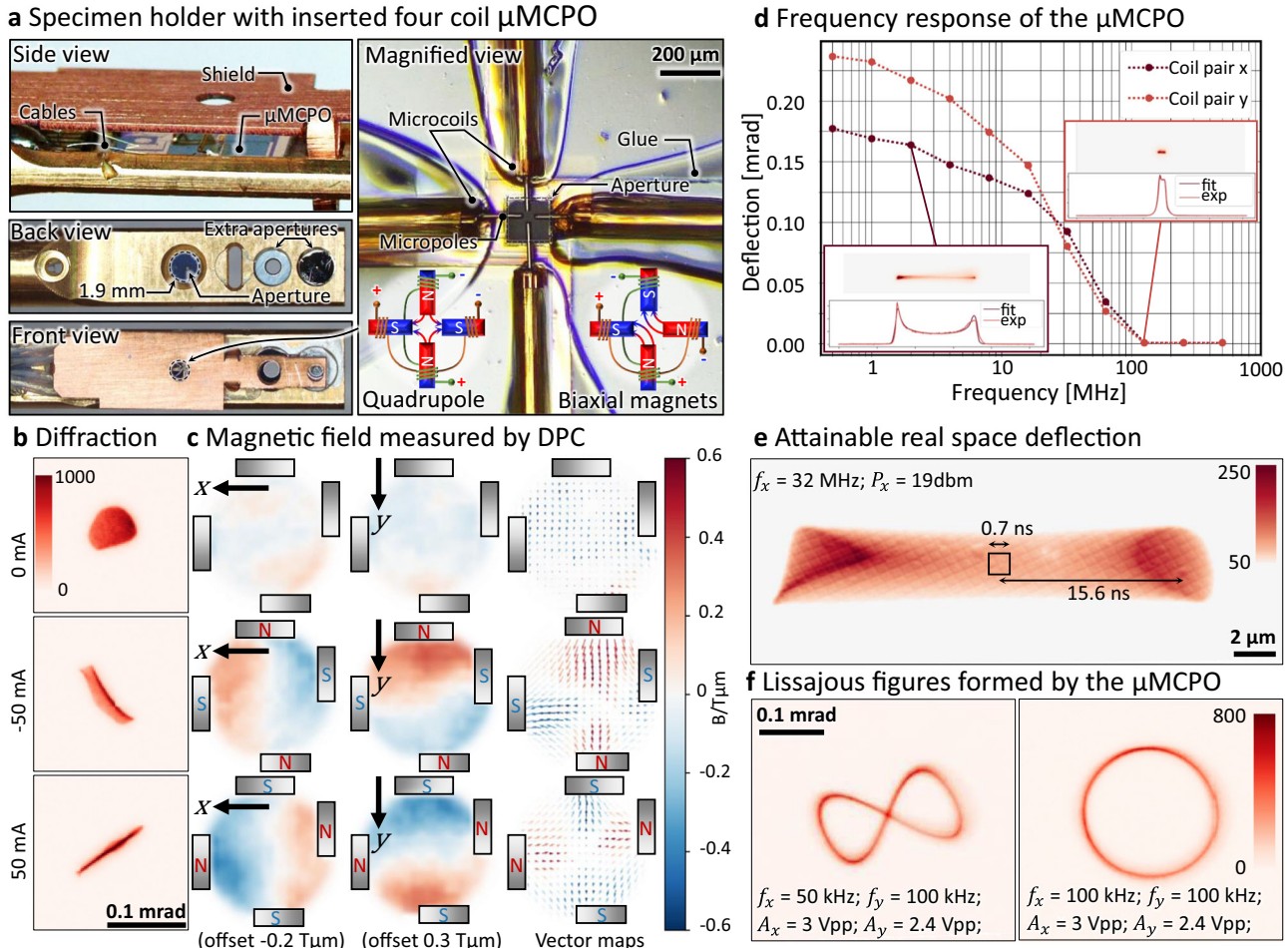

**Fig. 4 Quadrupolar micro-electron beam optics. a** On the left side: Aperture holder for the TEM with a custom shield ready for the insertion inside of the TEM column. On the right side: Magnified view shows the cores inserted inside of the micro-coils. Depending on the configuration of the contacts the set-up can work as a quadrupole or as a biaxial magnet. **b** Beam profile in the far field for parallel illumination of the µMCPOs in quadrupole configuration; all the coils are stimulated with a current of plus and minus 50 mA; the beam is thus focused in the xy and x(−y) axes, respectively. The same electrical stimulation of the sample is performed in **c** for the DPC measurement of the magnetic field: The first column depicts the x component of the magnetic field, the second column depicts the y component and the third column depicts the magnetic field vector. The reconstructed projected field values are offset linearly by the numbers indicated in the figure, to correct for an additional shift of the electron beam stemming from the misalignment of the poles with respect to the micro-coils. The colors in the vector plot represent the direction of the flux lines, to that end, the sine of the angle multiplied by two plus 90° is color mapped. **d** The amplitude of a slightly anharmonic sinusoidal deflection of the electron beam by a device in biaxial magnet configuration. The unidirectional deflection of the beam along x and y direction (perpendicular to the respective coils) is plotted as a function of driving frequency. The insets show the beam profile and the corresponding projection perpendicular to the oscillation direction used for the numerical evaluation at 2 and 128 MHz. **e** Attainable real space deflection at 32 MHz in the TEM's object plane for stroboscopic experiments. **f** The biaxial magnet can deflect the electron beam in x and y direction producing Lissajous figures (circle and the "∞" pattern) by simultaneous excitation currents with an appropriate mutual phase and frequency shift.

in a biaxial magnets configuration (i.e., not the alternating quadrupole configuration), we can use it as a scan generator with deflection angles of several mrads at tens of kVs accelerating voltages. To demonstrate the dynamic properties of the micro-coil devices we measured the frequency-dependent deflection of an electron beam that is extended in the micro-coil plane (C3 aperture of the TEM) and focused in the detector plane (i.e., detector is in the far field with respect to the micro-coil device). The insets in Fig. 4d show the intensity profile, which approximately follows the motion of a harmonic oscillator due to the sinusoidal excitation of one coil pair. The small asymmetry is due to field inhomogeneities introduced by the asymmetric poles and non-linearities of the magnetization switching (Fig. 3b) which can be compensated by a slightly anharmonic signal (see Supplementary Note 6 and Supplementary Fig. 5 for details). For

evaluating the deflection, the line profile is fitted to the probability density of an anharmonic oscillation (see Supplementary Fig. 4).

Accordingly, the frequency transfer of the whole assembly (cabling, holder, and µMCPO) currently yields a cut-off frequency of approximately 100 MHz (Fig. 4d). This is over one order of magnitude larger than previously reported microsized magnetic CPOs[17] and largely sufficient for applications such as ultrafast scan units in electron microscopes, which can reduce the impact of sample drift, and ultrafast beam blankers[37] for minimizing the irradiation dose absorbed by sensitive materials such as biological tissue and other organic samples.

Indeed, the expanded high frequency capabilities of beam shaping devices also enable the implementation of (ultra-)fast stroboscopic TEM imaging techniques. Figure 4e shows the cumulated intensity of a beam (illumination radius $r_s = 1.3\,\mu m$ in

the object plane) oscillating at 32 MHz with an amplitude of $A =$ 14.3 μm. Accordingly, a field of view (FOV) of 1 μm$^2$ is illuminated over 0.7 ns at a repetition rate of about 15 ns in this stroboscopic setup. The pulse width can be further decreased at the expense of a smaller FOV, e.g., pico-second pulse widths are attainable for FOVs in the nm$^2$ range. Currently, fast beam shaping and blanking in high energy CPO devices such as electron microscopes is exclusively facilitated by transversal cavities[1,38], which are, however, very bulky, costly and challenging to install. Last but not least, simultaneous excitation of all the coils of the biaxial magnet results into deflection in both $x$ and $y$ directions, which allows to produce a variety of Lissajous figures (Fig. 4f) in the detector plane (see Supplementary Note 7 for more details).

## Discussion

This letter shows an all-electrically driven modulator device for CPO realized via self-assembling 3D technique that opens the path for spatiotemporal electron beam modulation. On-chip micro-coils with diameters as small as 80 μm were fabricated on 6-inch wafer scale allowing for the parallel manufacturing of more than 1000 coils. We demonstrated that up to 4 coils can be fitted into the limited footprint available within a single chip, fitting into commercial TEM aperture holders. The micro-electromagnets are capable of generating transverse magnetic fields that effectively modulate electron beams. We showed a beam modulation up to 100 MHz, a quadrupole with a focal length of 46 mm, and a (half) vortex beam with OAM up to 10$^4$. Consequently, the μMCPO's performance, expressed in terms of deflection angle and focal length, is comparable with macroscopic magnetic devices that employ currents of several amperes to focus high energy charged particles. The limiting cutoff frequency of the assembly can be enhanced by improving the electrical configuration. Both the production yield and the electron optical performance can be eventually enhanced further by optimizing the self-assembly technique, in particular incorporating the fabrication of the soft-magnetic core, e.g., via electroplating (see Supplementary Note 8 and Supplementary Fig. 7), minimizing the misalignment of the poles, which currently induce significant parasitic aberrations.

The presented technology opens interesting avenues for CPO applications. That includes fast scanning units exploiting the small inductance (μH) of the devices, miniaturized modulators, sources of electromagnetic radiation[15], and very compact electron beam devices. These easily movable, parallelizable, and hence scalable, and inexpensive devices may be used in miniaturized sensors, electron microscopes[9,39], multibeam lithography devices, x-ray sources, charge neutralization tools, electron beam scrubbers, FEL[15,40], and inverse FEL accelerators[14], nuclear/spin resonance spectroscopy applications[25,26], amongst others. Stroboscopy techniques enable time resolved electron microscopy application, approaching the GHz regime allows to study dynamic micromagnetic phenomena at nanoscale such as ferromagnetic resonance of a magnetic vortex[41].

## Methods

**Fabrication steps**. First, a photo patterned SU8 mask is applied on a Si-wafer. A 40 nm Al$_2$O$_3$ layer is deposited on the opposite face by atomic layer deposition (ALD) in a FlexAL (Oxford Instruments Plasma Technology). The wafer undergoes a silanization process to promote the adhesion of the polymers. Afterwards, four polymer layers are photo-patterned successively on top of the Al$_2$O$_3$: sacrificial layer, polyimide, hydrogel, and polyimide. On top of the polymer stack Ti–Cu–Ti (5–300–5 nm) layers are sputtered (with a HZM-P4, Von Ardenne) and photo patterned by a lift-off process (with AZ 5214E, MicroChemicals). Afterwards, an aperture is etched on the Si-wafer with 0.0225 or 0.04 mm$^2$ area (depending on the design) with the Bosch Process (BP) in a PlasmaPro 100 ICP (Oxford Instruments Plasma Technology). The wafer is now diced into single samples in a SS10 (ACCRETECH (Europe)) dicing machine. Finally, the micro-coils are self-assembled, the soft magnetic wire is inserted and fixed in the coils, and then modified by

Focussed Ion Beam (FIB) milling. This step was done in a CrossBeam XB 1540 (Zeiss) and an NVision40 (Zeiss). In both devices Ga+ ions of 30 kV were used for milling with ion currents up to some 10 nA. The devices were also used for SEM imaging.

**Electric characterization details**. Automatized two-probe characterizations at wafer scale are performed in a probe station (Summit 12000; Cascade Microtech Inc., Beaverton, OR, USA) using a precision LCR Meter (E4980A Agilent Technologies Inc., Santa Clara, CA, USA). The resistance and inductance are measured at a frequency of 100 Hz. In the current break down test, the voltage ramps up to 5 V within 0.1 s through every coil by Agilent sourcemeter (Agilent Technologies Germany GmbH & Co. KG, Waldbronn, Germany). After the sweep, breakdown voltage and current were extracted from data by script searching for an abrupt change in flowing current.

**FEM Simulations**. Ansys electromagnetics is employed for FEM simulations in order to evaluate the attainable fields produced by the self-assembled micro-coils. The geometry of the micro-coils is closely matched to one of the real coils. The model for the simulated micro-coil has an outer diameter of 140 μm and a pitch between the "Swiss-role" layers of 0.6 μm. The cross section of the conducting layers is $100 \times 0.3$ μm. The resulting axial magnetic field for a DC of 100 mA in the vicinity of the conductors, is about 3–4 mT (c.f. Fig. 2c) Adaptive meshing was employed for the simulation within a simulation volume of $225 \times 225 \times 400$ μm.

**VSM characterization**. The magnetization behavior of the soft ferromagnetic micro-wires was analyzed with the Vibrating Sample Magnetometer (VSM) option inside the Physical Properties Measurement System (PPMS) from Quantum Design. The used wire had a length of 0.27 cm with a ferromagnetic core of 9.33 μm in diameter, as measured from SEM images. Various (minor) hysteresis loops were recorded without significant deviations from the initial complete loop shown in Fig. 2g. The corresponding axial flux density of the tip of the wire is about 0.7 T. The recorded magnetization curve was offset by −15.45 mT along the external field axis in order to center the curve. This procedure is correcting the hysteresis of the ferromagnets in the VSM's excitation coils. Note, however, that the external field acting on the micro-wire inside the VSM differs significantly from to the one produced by the μCPO devices. The former is homogeneous, while the latter is decreasing in dipolar fashion in the relevant region where the electron beam is interacting with the emanating field of the soft magnetic cores. Therefore, the absolute values are not directly comparable in these two experimental situations.

**TEM deflection experiments**. The μMPCO device is inserted in a Protochips Fusion Platform, which is then introduced into the object plane of a double-corrected TEM (FEI Titan$^3$ 80–300 TEM). The projected magnetic flux density within the μCPO elements is measured by means of a center-of-mass differential phase contrast (COM-DPC) technique inside the microscope that is operated in a specially configured low magnification STEM mode, which enables magnetic field free measurement conditions in the sample plane and a large field of view. The microscope is operated at 300 kV acceleration voltage, the deflected diffraction disks are recorded with an effective camera length of 13.16 m.

**HF measurements**. The HF transmission behavior of the μMCPO was evaluated ex situ with a 4294A Precision Impedance Analyzer (KEYSIGHT Technologies) network analyzer in a first step. The in situ HF response is directly analyzed via the deflection of the electron beam by the fields produced from a miniaturized biaxial magnet μMPCO (c.f. Fig. 4a, d). A R&S®SMR 27 Microwave Signal Generator excites two opposing coils simultaneously at frequencies ranging from 10 MHz to 1 GHz. A 33600A Series Trueform Waveform Generator (KEYSIGHT Technologies) excites the coils for smaller frequencies up to 120 MHz. The biaxial magnet μMCPO is placed in the C3 aperture position of a double corrected FEI Titan$^3$ 80–300 TEM, in order to analyze the attainable deflection in terms of deflection angle and real space deflection. The scanning corrector of the microscope was operated in a special transfer system mode in order to enhance the attainable deflection in real space in the specimen position. A prototype ThermoFisher aperture holder (based on ThermoFisher NanoEx i/v specimen holder) was modified with rg-178 impedance matched coaxial cables and SMA contacts in order to mount the miniaturized vector magnet in the C3 aperture position and to allow for a large HF passband.

## Data availability

The data that support the findings of this study are available from the corresponding author upon reasonable request.

## Code availability

Software codes that support the findings of this study are available from the corresponding author upon reasonable request.

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

## Acknowledgements

We greatly appreciate the support of the clean room team headed by R. Engelhard (IIN/IFW Dresden). We further thank S. Nestler, M. Bauer, and K. Leger (IIN/IFW Dresden) for the assistance with the deposition of $Al_2O_3$, the 6-inch mask fabrication, and the polymer synthesis, respectively. A special thank goes to S. Grundkowski and A. Müller-Knappe for laser cutting the aperture holder cover as well as D. Bieberstein (IKM/IFW Dresden) for FIB milling support. We thank Ulrike Nitzsche for technical assistance with the FEM simulation setup. A.L. acknowledges funding from the European Research Council (ERC) under the Horizon 2020 research and innovation program of the European Union (grant agreement no. 715620). O.G.S. acknowledges support by the German Research Foundation DFG (Gottfried Wilhelm Leibniz Prize granted in 2018, SCHM 1298/22-1). D.K. acknowledges support by the German Research Foundation DFG (KA5051/1-1 and KA 5051/3-1), as well as by the Leibniz association (Leibniz Transfer Program T62/2019).

## Author contributions

F.K. and R.H. contributed equally to this work. O.G.S., D.K., B.B., and A.L. conceived the idea. F.K. performed FEM simulations, VSM measurements and participated in the design of the samples with R.H., who did the fabrication of the samples. D.K. and D.D.K. developed polymeric self-assembly materials and process. R.H., E.E., and P.L. developed the stripes-based directional self-assembly process design. C.B., D.D.K., T.K., and A.M. helped in the technological development and tuning of the polymer technology. T.K. did the synthesis of the polymers and assisted during manufacturing. D.D.K. with the help of R.H. did the electrical characterization of the structure on the wafer and its analysis. A.M. diced the wafers. S.B. and F.K. operated the FIB. D.K. with D.D.K has accomplished high frequency cabling and characterization of the aperture holder. D.K. with R.H. did the installation and wire-bonding of the sample to the customized TEM aperture holder. F.K. with help of R.H. did the TEM measurements and analysis of the data, A.T. contributed to the high-frequency beam deflection experiments and analysis. R.H., F.K., D.K., and A.L. wrote the manuscript with input from all authors and supported experimental design. O.G.S., D.K., and A.L. supervised the work. All authors participated in data analysis and discussion.

## Funding

## Competing interests

The authors declare no competing interests.
