## [Peer Review File · Nature Communications]

Tailoring electron beams with high-frequency self-assembled magnetic charged particle micro opticsREVIEWER COMMENTS

Reviewer #1 (Remarks to the Author):

I have read with interest The paper of Huber et al. It is full of very interesting physics and potentialities. I think this will be a seminal work for the seminal for many activities of the group in future.

In general the article is well written in terms of clarity. The general concept is clear: with a new self-assembled “origami” technique the author can drive 4 soft ferromagnets.

The numbers are impressive: band pass at 100 MHz, 0.2T , 10^4 quanta of flux.

These numbers are sufficient to make a miniaturized device that can compete with the more bulky approaches to beam shopping at high frequency.

In this sense it would be worth to mention, beyond ref 34 also

X. Fu et al . Science Advances DOI: 10.1126/sciadv.abc3456

which is the other main player at the moment on RF cavities.

For what I see the main application is, therefore, in ns scale time resolved microscopy and without the help of cooling mechanisms that is instead necessary in radio frequencies.

In this sense the most surprising fact is that there is no need of a ohmic coupling to cut reflection and detailed study of maxwel equation in cavity.

If we see the beam shaping aspects I would like to highlight that

1) an half vortex is not a vortex. This can seem a semantic problem but for whatever reason one may want a vortex, having half of that makes it completely useless. If possible smooth this claim a bit, but the discussion is overall quite honest. Still I don't think the same OAM can be reached with a smaller tip.

The current largest vortex are $L=1000$ (Mafakheri Applied Physics Letters 110 (9), 093113)

And a superimposed high order with $L=5000$

B. J. McMorran, Phil. Trans. R. Soc. A 375, 20150434 (2017)

Please update the references.

2) The Lissajous pattern that are shown are quite simple and low order and yet blurry. This makes me think that there is quite a substantial phase jitter. If the author want to claim potentially very high

frequency sub –ns application this jitter would be a big obstacle for real application because the delay between pump and probe would be scattered in time.

Can the author quantify this problem and the potential impact .

My guess is that the problem arises from the claimed non linearity effects.

3) There are recent electrostatic miniaturization in electron optics like

Tavabi Phys. Rev. Lett. 126, 094802

Consider also a few of these works

Zheng et al. 10.1103/PhysRevLett.119.1748011

And theoretical proposals for magnetic phase elements

Pozzi Ultramicroscopy 113287 (2021)

Overall I consider the work of very good quality and interest after the minor point raised above are considered.

Reviewer #2 (Remarks to the Author):

The paper entitled "Tailoring electron beams with high-frequency self-assembled magnetic charged particle micro optics" by R. Huber et al, describes an innovative method for the fabrication of miniature electron optics components. This new method is based on the use of microfabrication techniques such as photolithography combined with focused ion beam preparation techniques. These miniature systems have been installed inside a TEM in order to test their ability to generate vortex beam, deflectors but also focusing systems such as magnetic quadrupole.

Overall the paper is very well written, the results are simply impressive and very well documented thanks to the extensive supplementary informations.

I find the paper very suitable for the journal Nature communications and, in my opinion, it could be published as is.

In order not to conclude this report without a scientific discussion, I am reporting below the questions I would like to discuss with the authors. Their answers might also clarify some specific points for future readers:

- I guess that the glass deposit on the surface of the poles is used to insulate it from the electric contact of the coil which are surrounding the wire ? If so, I think it should be specified in the paper. If not, then how do you carry out properly this insulation?

- So, when shaping the surface of the pole with the FIB, you remove a large part of this glass material? Have you ever seen charging problems due to the remaining glass coated surface or due to redeposition phenomena?

- It is also well known that the magnetic properties of soft materials depend on the machining methods. Mumetal is an extreme case where machining is extremely complex if you want to keep the properties intact. Could you comment on the evolution of the magnetic properties before and after FIB milling? You spoke about Ga implantation for instance but how do you think it will affect the properties (remanence, saturation, ...). To answer properly to this question, it would be ideal to show two hysteresis curves before and after milling.

-Why was the surface of some nanowires machined in the shape of a triangle? In order to have the best homogeneity of the magnetic field along the electrons trajectories I would have naively left large surfaces...

-Do you think it could be possible to use your methods to prepare a round lens type ? for example by "drilling" a hole along the nano wire, previously cut in between the two coils as in the case of deflectors. The system will then have to be rotated by 90 degrees allowing the electrons beam going through the wire hole.

- Concerning quadrupoles, you succeed to measure their paraxial properties. How do you consider the effect of fringing fields? Indeed, this type of system is very sensitive to fringing fields, and in general magnetic shields are used to confine these fields located on the two quad edges so that the paraxial properties could be kept as close as possible to those calculated in the sharp cut-off approximation.

- It is also well known that the fringing fields have a strong effect on the intrinsic aberrations of quad systems (especially aperture ones). In your case the aberrations are clearly parasitic ... , however it would be nice to have a comment on the fringing fields distribution. Indeed, due to the very short total length of your quadrupole, these two fields effects on the primary aberrations of the type (x/aaa), (x/abb) will be considerable (contribution to 50% or more in my opinion).

-Finally, you mention potential applications for ion beams. What exactly do you have in mind? Indeed, I would be very careful with this statement because it is well known that magnetic systems are not interesting for ions, except in extreme energy situations. A comment or example of application should be interesting. If not, I would have remove this statement.

These points are only minor questions and their answers would make the referee that I am happy, but in my opinion, they are not mandatory to allow a publication in Nature Com.

Congratulations again to all the authors for this great work.

Reviewer #3 (Remarks to the Author):

This paper describes 3D microelectromagnets for manipulating charged particle beams. The devices can produce fields up to 100 mT, and they can operate at frequencies up to 100 MHz due to their low inductance. As a demonstration, a 300 kV electron beam was modulated. For example, orbital angular momentum was imparted on an electron beam, with the motivation being that such a beam could be used as an experimental probe. There were also demonstrations of beam steering with a dipole field and beam focusing with a quadrupole field. Specific questions are given below.

Questions/comments:

- In your introduction, you mention “electromagnets are typically macroscopic (super-)conducting coils, which cannot generate swiftly changing magnetic fields, require active cooling to dissipate heat...” The way your coils are designed, it seems that heat dissipation would also be a challenge, such as what you show in figure 2f. It seems this comment about heat dissipation also applies to your work.
- The coils are made in a bulk process, which is great. However, the hand assembly for the quadrupoles limits their impact. Similarly, hand inserting a soft magnet wire and using FIB to pattern it will limit the ways in which these devices can be used in general. If there are ways to replace some of these serial and/or hand assembly processes with bulk processes, that would enhance the impact of the work.
- For your impact statement, you mention these are the “the first on-chip micro-sized magnetic charged particle optics (uMCPOs) realized via a self-assembling micro-origami process.” So, they aren’t the first uMCPOs, but they are the first uMCPOs realized with the self-assembling origami process. To make this impact statement more robust, you could add more concrete details about why the self-assembling

origami process is better than other methods for making small magnets (e.g., higher energy density devices? Different types of devices on a single wafer? I don't know the answer to this, but I'm sure you do).

- The > 90% yield on the coil fabrication process is impressive.
- For the deflection angle of the beam (and the field you infer from the angle), why is it so asymmetric (-109 mT vs 67 mT)? You mentioned some possible technological reasons. It would be interesting to make some basic calculations to see how much of a technological variance (e.g., how much wire inhomogeneity) would be required to generate this asymmetry.
- You achieve a stated quadrupole gradient of 800 T/m using a sharp cutoff approximation. I'm guessing you mean you're assuming there aren't fringe fields and that you have a uniform gradient in the 20 μm thickness of the wire. Is this approximation valid? If I understand correctly, the spacing between your poles is 30 or 40 microns, and y =the wire thickness is 20 μm . With the gap bigger than the wire thickness, it would seem that you have substantial fringing fields. You'll still get focusing of course, but using a sharp cutoff approximation doesn't make sense.
- You acknowledge the misalignment in the quad. Generally speaking, people that use quadrupoles would require a field that is much more well controlled than one with the misalignment. You mentioned being able to compensate with different currents. It would be useful to show to what degree you could fix the quadrupole field with currents, and how much you would need to improve alignment.

Dear Reviewers,

thank you very much for taking the time and effort reviewing our paper and providing very helpful remarks and critics. We tried to diligently respond to all of them, see detailed answers below. Generally, all important changes in the revised manuscript have been marked red.

Reviewer #1 (Remarks to the Author):

I have read with interest the paper of Huber et al. It is full of very interesting physics and potentialities. I think this will be a seminal work for the seminal for many activities of the group in future. In general, the article is well written in terms of clarity. The general concept is clear: with a new self-assembled “origami” technique the author can drive 4 soft ferromagnets. The numbers are impressive: band pass at 100 MHz, 0.2T, 10^4 quanta of flux. These numbers are sufficient to make a miniaturized device that can compete with the bulkier approaches to beam shopping at high frequency. In this sense it would be worth to mention, beyond ref 34 also X. Fu et al . Science Advances DOI: 10.1126/sciadv.abc3456, which is the other main player at the moment on RF cavities.

A1.1: We thank the referee for the positive evaluation of our work and bringing to our attention the impressive work on RF cavities, which is now cited in the revised version, when discussing ultrafast beam blanking.

For what I see the main application is, therefore, in ns scale time resolved microscopy and without the help of cooling mechanisms that is instead necessary in radio frequencies. In this sense the most surprising fact is that there is no need of an ohmic coupling to cut reflection and detailed study of maxwel equation in cavity.

A1.2: We see two main application of the μ coil devices as far as electron optics is concerned: (A) ns-scale time resolved microscopy as pointed out by the referee and (B) miniaturized electron optical instruments. To overcome the current 100 MHz limit encountered in (A) further developments are underway, which particularly address the HF optimization of the chips (i.e., by incorporating better shielded planar wave guides) and the wiring. We agree with the referee that this (including detailed electromagnetic modeling of the setup) is required to reach the GHz regime.

If we see the beam shaping aspects, I would like to highlight that 1) a half vortex is not a vortex. This can seem a semantic problem but for whatever reason one may want a vortex, having half of that makes it completely useless. If possible, smooth this claim a bit, but the discussion is overall quite honest. Still I don't think the same OAM can be reached with a smaller tip.

A1.3: We fully agree with the referee that the current shape of the „vortex“-beam is not well suited for application (e.g., electron optical tweezer or out-of-plane magnetization probe). Note, however, that a previously reported vortex beam generator based on a ferromagnetic nanowire (Béché et al., Nature Physics 10, 26-29, 2014 Fig. 3 and Suppl. Fig. 2) suffered from very similar imperfections and we attribute them to two main causes: (I) Charging of the microwire (including the glass coating) in the electron beam and (II) magnetic stray fields emerging from the sides of the wire (in particular at the tapered regions). While the first effect produces a marked asymmetry of the vortex (see, e.g., Pozzi et al., Ultramicroscopy (2017), doi: 10.1016/j.ultramic.2017.06.001), the second restricts the azimuthal range of the deflection and hence the arc angle. We studied (and tried to improve) these aspects by FIB preparing a lamella-like tip (in order to reduce field lines emerging from the sides, see Fig. 1)

and by coating a tip with Au (not shown). Both measures improved the vortex shape. Note, however, that significant deviations from a real vortex persist, presumably due to the pole shadow and remaining magnetic field lines emerging from the sides of a non-homogeneously magnetized poles (indeed, our vortex state microwires can sustain complicated tilted vortex or flux closure domains in the tip region). We added a small supplement (Supplementary Note 4) containing the comparison of “vortex” beams generated by the tapered and the lamella-like tip including the following Fig. 1 and the above discussion.

Fig. 1 Comparison of tapered (a,b) and lamella-shaped (c,d) electron vortex beam generating μ coil devices. The “half vortices” generated by the lamella shaped pole exhibits an increased arc in the far field covering a larger angle ($>180^\circ$) as compared to the tapered microwire tip ($<180^\circ$).

The current largest vortex is $L=1000$ (Mafakheri Applied Physics Letters 110 (9), 093113)
 And a superimposed high order with $L=5000$
 B. J. McMorran, Phil. Trans. R. Soc. A 375, 20150434 (2017)
 Please update the references.

A1.4: Done

2) The Lissajous pattern that are shown are quite simple and low order and yet blurry. This makes me think that there is quite a substantial phase jitter. If the authors want to claim potentially very high frequency sub-ns application this jitter would be a big obstacle for real application because the delay between pump and probe would be scattered in time. Can the author quantify this problem and the potential impact? My guess is that the problem arises from the claimed nonlinearity effects.

A1.5: We thank the referee for the excellent question. We agree with the referee that phase jitter will be a problem when approaching the GHz regime. Note, however, that the inhomogeneous blur observed in our Lissajous patterns is of different origin as demonstrated in Fig. 2 below. The position-dependent blurring of the beam, while preserving the total intensity (i.e., area under the cross section) can be attributed to an inhomogeneous magnetic field distribution / deviations from the ideal superposition of two perpendicular homogeneous

fields, generating an additional focusing effect, visible as a position dependent blur. This comment including the figure has been added to Supplementary Note 7 in the revised version.

Fig. 2: Cross section analysis of circular Lissajous pattern depicted in upper right corner.

3) There is recent electrostatic miniaturization in electron optics like

Tavabi Phys. Rev. Lett. 126, 094802

Consider also a few of these works

Zheng et al. 10.1103/PhysRevLett.119.174801

And theoretical proposals for magnetic phase elements

Pozzi Ultramicroscopy 113287 (2021)

A1.6: We again thank the referee for this literature suggestions and included first two of the above references in the introduction. We did not include the third one because (A) we are restricted in the total number of references due to formatting guidelines and (B) this theoretical work on OAM sorters does not really fit to our work.

Overall I consider the work of very good quality and interest after the minor point raised above are considered.

Reviewer #2 (Remarks to the Author):

The paper entitled "Tailoring electron beams with high-frequency self-assembled magnetic charged particle micro optics" by R. Huber et al, describes an innovative method for the fabrication of miniature electron optics components. This new method is based on the use of microfabrication techniques such as photolithography combined with focused ion beam preparation techniques. These miniature systems have been installed inside a TEM in order to test their ability to generate vortex beam, deflectors but also focusing systems such as magnetic quadrupole. Overall, the paper is very well written, the results are simply impressive and very well documented

thanks to the extensive supplementary information.

I find the paper very suitable for the journal Nature communications and, in my opinion, it could be published as is.

A2.1: We thank the referee for the positive remarks.

In order not to conclude this report without a scientific discussion, I am reporting below the questions I would like to discuss with the authors. Their answers might also clarify some specific points for future readers:

- I guess that the glass deposit on the surface of the poles is used to insulate it from the electric contact of the coil which are surrounding the wire? If so, I think it should be specified in the paper. If not, then how do you carry out properly this insulation?

A2.2: This is a misunderstanding. The glass coating of the magnetic microwires is an essential part of their fabrication process as it allows to impose strain and hence a magnetostrictive force promoting an ultrasoftmagnetic longitudinal vortex configuration within the microwire. As a side effect it also provides the electrical insulation (for free) preventing any shortcut of the microcoils due to softmagnetic filling. These points have been clarified in the revised text.

- So, when shaping the surface of the pole with the FIB, you remove a large part of this glass material? Have you ever seen charging problems due to the remaining glass coated surface or due to redeposition phenomena?

A2.3: Depending on how the poles are shaped also the glass is removed as can be seen in the new Supplementary Figure 4. The evaluation of the charging problems, however, is not fully completed yet. Currently, we, e.g., observe that charging plays a role in the deformation of the vortex beams (e.g., we observed different fields upon Au coating of the poles), however, it is difficult to disentangle these effects from magnetic contribution, which are also modified by the presence or removal of the glass coating (partly lifting the strain and hence the magnetostriction).

- It is also well known that the magnetic properties of soft materials depend on the machining methods. Mumetal is an extreme case where machining is extremely complex if you want to keep the properties intact. Could you comment on the evolution of the magnetic properties before and after FIB milling? You spoke about Ga implantation for instance but how do you think it will affect the properties (remanence, saturation, ...). To answer properly to this question, it would be ideal to show two hysteresis curves before and after milling.

A2.4: See also previous comment. The impact of fabrication on the magnetic properties of the wire is a very important point, and under further investigation by us. Note, however, that ultimately (at least to our understanding, see also response A3.2 to Ref. 3) the soft magnet filling should be integrated in the lithographic fabrication process (we currently evaluate electroplating of Py / related materials, see Fig. 3 further below) in order to achieve a better alignment of the poles (and hence less parasitic aberrations) with higher production yield (the manual insertion of the microwires is error prone). After this technology change, specific properties of the soft magnet filling will change significantly (because magnetic ground state of soft magnet filling will not necessarily be of longitudinal vortex type (that depends on the thickness to diameter radius of the filling, amongst others)), rendering a thorough study of the glass-coated microwires less important for further development, to our point of view.

-Why was the surface of some nanowires machined in the shape of a triangle? In order to have the best homogeneity of the magnetic field along the electrons trajectories I would have naively left large surfaces...

A2.5: Our initial idea was to channel and concentrate the magnetic field of the microwire while reducing the shadow, and hence to produce the highest possible OAM in the vortex beam. While the field is indeed very large, the results, however, indicate that this geometry has severe drawbacks, notably the strong deviation of the emerging field from an ideal Dirac string (see response A1.3 to referee 1 and new Supplementary Note 4). The referee's "naive" suggestion of a flat cylinder shape was therefore indeed used for the dipole and quadrupole devices throughout, where beam blocking is not an issue (electron beam is passing in between the poles). Indeed, further analysis of the fields (not shown here) indicate that the radial shape of the microwires and the thereby induced profile of the stray fields is helpful to approximate the ideal hyperbolic pole shapes of the quadrupole reducing aberrations. Consequently, further optimization of the pole shape is a useful strategy for improving the performance of the devices after the problem of parasitic aberrations due to misalignment of the poles has been resolved by direct deposition of soft magnet core.

-Do you think it could be possible to use your methods to prepare a round lens type? for example by "drilling" a hole along the nano wire, previously cut in between the two coils as in the case of deflectors. The system will then have to be rotated by 90 degrees allowing the electrons beam going through the wire hole.

A2.6: We don't completely understand the question and therefore apologize if the response is misleading. When we started our studies, we indeed initially thought about round lenses, where the beam is traveling along the microcoil axis. However, it quickly turned out that the second order focusing effects of round lenses is too weak (or would require too large a magnetic field, which cannot be produced with our coils) to be of any use.

- Concerning quadrupoles, you succeed to measure their paraxial properties. How do you consider the effect of fringing fields? Indeed, this type of system is very sensitive to fringing fields, and in general magnetic shields are used to confine these fields located on the two quad edges so that the paraxial properties could be kept as close as possible to those calculated in the sharp cut-off approximation.

A2.7: This is again an excellent question touching various charge particle optics details, some of which we don't yet understand completely yet. First of all, considering the scope of this paper, the proof-of-principle status of the devices, and the dominating parasitic aberrations due to misalignment of the poles, we were generally satisfied with a paraxial description of our system, reproducing the observed paraxial properties such as the focal length of the quadrupoles. Moreover, we observed that due to the small extension of the fields along the beam direction, the devices (e.g., deflector or quadrupole) may be treated as thin devices (i.e., thin deflector, thin quadrupole), where integrals of the axial fields along the beam direction determine the paraxial characteristics (described by one cardinal element, e.g., the focal length (or two focal lengths in a nonsymmetric quadrupole)) completely. For instance, according to (Hermann Wollnik, "3 - Quadrupole Lenses", in *Optics of Charged Particles* (Academic Press, 1987), pp. 48-88.), the thin element approximation yields good results for quadrupoles with $kl < 0.8$, with $k = \sqrt{\frac{eC_{qp}}{m_e v}}$. Our miniaturized quadrupoles have a kl of about 0.03, which renders the thin element approximation well valid. In that sense the sharp cut off is merely an

approximation of that projection integral, which can be made exact if choosing the correct effective width of the cut-off region. The corresponding phrase in the manuscript has been reformulated to reflect the above discussion.

Having said this, the question of the impact of fringing fields on aberrations of the microcoil devices and consequently the usability of sharp cut-off approximation in calculations of aberrations and design efforts to mitigate fringing fields needs to be considered in the further development of the devices as the aberrations will be dominantly determined by the fringing fields (and thin device approximation is of little use when it comes to aberrations). Note, however, the fact that fringing fields affect aberrations in itself is not necessarily bad (e.g., c.f. efforts to reduce intrinsic aberrations of, e.g., sector magnets, by shaping fringing field). Furthermore, application of sharp cut-off approximation in simulations is not necessarily simplifying them. Indeed, the additional effort to incorporate additional boundary conditions at the sharp field drops is considerable, and frequently reported sharp cut-off calculations in literature are indeed continuous field calculations with very steep (but continuous) field changes. From our preliminary considerations of the fringing field impact on the quadrupole device we note that a better approximation of the field distribution along z of our devices is indeed the bell-shaped model, which also has the advantage that closed analytic evaluation of aberration integrals in terms of circular and hyperbolic functions is possible (e.g., Hawkes (1966), *Quadrupole Optics*, Springer). A detailed consideration of the aberrations is, however, beyond the scope of this paper for the initially noted reasons.

- It is also well known that the fringing fields have a strong effect on the intrinsic aberrations of quad systems (especially aperture ones). In your case the aberrations are clearly parasitic ... , however it would be nice to have a comment on the fringing fields distribution. Indeed, due to the very short total length of your quadrupole, these two fields effects on the primary aberrations of the type (x/aaa), (x/abb) will be considerable (contribution to 50% or more in my opinion).

A2.8: See previous comment for impact of fringing fields. The referee correctly alludes to the dominating influence of the parasitic aberrations (in contrast to intrinsic ones), which are dominating our devices currently. Unfortunately, the situation in the literature concerning a systematic classification of parasitic aberrations of quadrupoles and quadrupole multiplets is disappointing (see, e.g., Kasper and Hawkes, *Principle of Electron Optics 2*, Chap. 39, and follow up literature). In particular, we could not find practical expressions allowing to determine fabrication tolerances, which would allow a suppression of parasitic aberrations (e.g., below the first order chromatic and second order aperture aberration inherent to a symmetric quadrupole). Consequently, a numerical evaluation of charge particle optics including aberrations of the uMCPOs devices including geometric distortions (i.e., misalignment of poles, etc.) has to be carried out, which was, however, beyond the scope of the paper.

-Finally, you mention potential applications for ion beams. What exactly do you have in mind? Indeed, I would be very careful with this statement because it is well known that magnetic systems are not interesting for ions, except in extreme energy situations. A comment or example of application should be interesting. If not, I would have remove this statement.

A2.9: We agree with the referee that electrostatic optics is the state of the art for ion optics. Our initial idea was to employ the devices in miniaturized ion beam systems, which is, however, a bit far-fetched. We removed the statement.

These points are only minor questions and their answers would make the referee that I am happy, but in my opinion, they are not mandatory to allow a publication in Nature Com.

A2.10: Here we disagree with the referee. All mentioned points are very important and shape our ongoing work aimed at improving the devices and applying them in various applications. We hope that our responses satisfied the referee.

Congratulations again to all the authors for this great work.

Reviewer #3 (Remarks to the Author):

This paper describes 3D microelectromagnets for manipulating charged particle beams. The devices can produce fields up to 100 mT, and they can operate at frequencies up to 100 MHz due to their low inductance. As a demonstration, a 300 kV electron beam was modulated. For example, orbital angular momentum was imparted on an electron beam, with the motivation being that such a beam could be used as an experimental probe. There were also demonstrations of beam steering with a dipole field and beam focusing with a quadrupole field. Specific questions are given below.

Questions/comments:

- In your introduction, you mention “electromagnets are typically macroscopic (super-)conducting coils, which cannot generate swiftly changing magnetic fields, require active cooling to dissipate heat...” The way your coils are designed, it seems that heat dissipation would also be a challenge, such as what you show in figure 2f. It seems this comment about heat dissipation also applies to your work.

A3.1: In the fabricated devices, the active coil is the most vulnerable part to current overdrive because of two reasons: (1) the copper strip that forms the coil has a small width and thickness, and (2) the coil is not in contact with the Si substrate. Because of this, heat dissipation is indeed a challenge when it comes down to an electrical current flowing through a coil, and heat dissipation is currently the limiting factor preventing currents larger than 100 mA and hence larger field strengths. We found that the heat dissipation directly scales with the thickness of the Cu conduction lines. Consequently, increasing the thickness of the lines would allow even larger currents. Note, however, that the coil radius also scales with the Cu line thickness. Thus, increasing the maximal current goes along with larger coils, which reduces the field strength on the coil axis. Hence, optimization of Cu line thickness is required to find an optimum with respect to maximum magnetic field. This has been done to some extent leading to a slightly larger thickness of ~250 nm in the next generation devices (for the paper 200 nm was employed, this information has been added to the revised manuscript). For the devices reported in the paper, we electrically tested the coils before employing them and choose the ones that withstand the 100 mA limit for further processing. By either employing pre-tested coils or improved coils, the employed devices have enough passive heat dissipation to withstand the required electrical currents without the need of active cooling. We added the above information on the conducting layer thickness and pretesting to the manuscript.

- The coils are made in a bulk process, which is great. However, the hand assembly for the quadrupoles limits their impact. Similarly, hand inserting a soft magnet wire and using FIB to pattern it will limit the ways in which these devices can be used in general. If there are ways to replace some of these serial and/or hand assembly processes with bulk processes, that would enhance the impact of the work.

A3.2: We fully agree with the referee and efforts going into this direction are underway (see Fig. 3). Specifically, we are currently trying to incorporate electroplating of the soft magnetic bars into the lithographic fabrication process, which would largely reduce misalignment of the poles, increase homogeneity of devices and fabrication speed. While plating and shaping of the poles have been successful, we still struggle to reproduce the ultrasoft magnetic properties of the hand-inserted microwires (sustaining longitudinal vortex ground state), which requires further optimization of the material composition, the geometry of the filling and electroplating parameters (or different magnetization procedures, where minor hysteresis loops are employed, ...). We added a comment on the improved fabrication process in the outlook section (which has been otherwise reformulated to also highlight the importance of parasitic aberrations better) and added a Supplementary Note 8 containing electroplating details including Fig. 3.

Fig. 3: **a**, Schematic of the permalloy (Py) bar deposition on the planar chip design in the dipole configuration before self-assembly (roll up of the coils). **b**, Image of the wafer including Py bars with a zoom in of the Py bars over the aperture position. The undesired joint is opened in the next step by a short current pulse.

- For your impact statement, you mention these are the “the first on-chip microsized magnetic charged particle optics (uMCPOs) realized via a self-assembling micro-origami process.” So, they aren’t the first uMCPOs, but they are the first uMCPOs realized with the self-assembling origami process. To make this impact statement more robust, you could add more concrete details about why the self-assembling origami process is better than other methods for making small magnets (e.g., higher energy density devices? Different types of devices on a single wafer? I don’t know the answer to this, but I’m sure you do).

A3.3: Indeed there has been only one other previous realization of microsized magnetic charged particle optics (Ref. J. Harrison *et al.*, High-gradient microelectromechanical system quadrupole electromagnets for particle beam focusing and steering, PR Special Topics – Accelerators and Beams 18 (2015), which is cited in our work): In this impressive work, the whole fabrication of the 3D structure was done lithographically, which particularly included rectangular coils and the soft magnetic cores. The conducting elements had dimensions of $100\ \mu\text{m} \times 80\ \mu\text{m}$ (on the bottom element) and $20\ \mu\text{m} \times 80\ \mu\text{m}$ (on the top element). A maximum current of 1.5 A was reported (without stating whether DC or pulsed), which corresponds to a current density of about $0.2\ \text{mA}/\mu\text{m}^2$ and $0.9\ \text{mA}/\mu\text{m}^2$, respectively. The current density in the coils fabricated in our work is significantly higher at about $6.6\ \text{mA}/\mu\text{m}^2$. We therefore think

that the employed micro origami process allows conduction layers of higher quality as they can be deposited in a single step, which ensures a low resistance and thus minimizes Joule heating. The size of the crucial parts of the optical elements are about one magnitude larger than in our work, e.g., the pole gap and the device length along the optical axis are 600 μm and about 180 μm , respectively, while they are 50 μm and 20 μm in our work. The most important electron optical specification reported in this work is the quadrupole strength, which is 57.6 T/m. This is again one order of magnitude below the value we obtained, which is 800 T/m, directly translating in significantly higher optical power of our quadrupole devices. The highest reported transmitted frequency in this work is 5 MHz, while we could demonstrate a transmission of an electric signal at 100 MHz. From that one could summarize that our uMCPOs devices perform significantly better in key areas. One should, however, also note that the devices reported by Harrison et al., currently seem to have the advantage to be fabricable completely lithographically without requiring additional insertion of magnetic microwires. As some of the details of these reported quadrupole devices are somewhat unclear for us (e.g., was the maximum reported current achieved in DC or pulsed mode) we were, however, reluctant to include such a detailed comparison of figure of merits (eventually evaluating previous work too negatively) in our paper and just slightly updated the manuscript by adding the comparison of HF performance and quadrupole strength at the appropriate positions.

- The > 90% yield on the coil fabrication process is impressive.
- For the deflection angle of the beam (and the field you infer from the angle), why is it so asymmetric (-109 mT vs 67 mT)? You mentioned some possible technological reasons. It would be interesting to make some basic calculations to see how much of a technological variance (e.g., how much wire inhomogeneity) would be required to generate this asymmetry.

A3.4: We thank the referee for the excellent question. Indeed, a thorough evaluation of different uMCPO devices of the same and different batches revealed a systematic bias, which showed some dependency on the FIB fabrication of the tips (i.e., shape). This suggests that the origin of the observed asymmetries (biases) is the magnetic configuration of the tips (and the wire). Indeed the longitudinal vortex magnetic texture within the wire in itself shows a small hysteric behavior due to the core (while the in-plane vortex spins tilt out of plane linearly the core cannot be tilted for topological reasons, it is rather pushed out of the wire in a complicated way during magnetization) and hence an apparent bias of our observed magnetization curve representing one branch of the hysteresis. Superimposed on those effect is the complicated and as of now not understood magnetization texture in the tip itself. Tomography studies of nanowire tips, e.g. Wolf et al. (2015) (<https://doi.org/10.1021/acs.chemmater.5b02723>), suggest the presence of flux closure domains, which may also contribute to some bias. Last but not least the whole situation is further complicated by the fact that the microwire is only partially immersed in the magnetic field of the coil. Hence, a linear, hysteresis-free and unbiased magnetization behaviour present in fully immersed wires may be modified upon half-immersion in the field. We therefore believe that the question of the bias may be only resolved by further magnetization studies of differently immersed microwires, and corresponding changes to the pole geometry. Note, however, that from a operational point of view, the application of a small bias in the range of 10 mA is not problematic.

- You achieve a stated quadrupole gradient of 800 T/m using a sharp cutoff approximation. I'm guessing you mean you're assuming there aren't fringe fields and that you have a uniform gradient

in the 20 μm thickness of the wire. Is this approximation valid? If I understand correctly, the spacing between your poles is 30 or 40 microns, and y =the wire thickness is 20 μm . With the gap bigger than the wire thickness, it would seem that you have substantial fringing fields. You'll still get focusing of course, but using a sharp cutoff approximation doesn't make sense.

A3.5: For the thin quadrupoles considered here, we may use the weak / thin quadrupole approximation, where both principal planes coincide and the paraxial properties of the quadrupole may be described by one quantity, e.g., the focal length. The latter is given by $p_0/(efC_{Qpdz})$. This integral was computed assuming a rectangular field and an approximation of the quadrupole length l by the diameter of the microwire. Concerning the paraxial properties this assumption is indeed superfluous and we agree with the referee that the rectangular field is not a good approximation. Indeed, a bell-shaped field is better suited (and yields similar results within the error of our measurement method, when choosing the width parameter appropriately). See response A2.7 to referee 2 for a discussion of the field shape on aberrations. We modified the text taking into account the above discussion.

- You acknowledge the misalignment in the quad. Generally speaking, people that use quadrupoles would require a field that is much more well controlled than one with the misalignment. You mentioned being able to compensate with different currents. It would be useful to show to what degree you could fix the quadrupole field with currents, and how much you would need to improve alignment.

A3.6: Indeed, the misalignment of the poles and the ensuing aberrations are the most critical drawback of our devices at this point (see A3.2 for mitigation efforts by improved fabrication). The possibility to correct part of the misalignment by individually driving the coil currents (which is currently not possible because the limited space on the chips (commercial Protochips / FEI aperture holders) and limited number of contacts allows only excitation of pairs of coils, a more elaborate chip design including a dedicated aperture holder fitted with more than 4 feed-throughs is under preparation) concerns the suppression of focal length differences in the focusing and defocusing plane and the suppression of dipole terms (i.e. deflections) with respect to the central axis of the quadrupole device. In other words, the leading order effect of the poles' misalignment is a shift of the optical axis (i.e., the axis where the electron passes straight) with respect to the central axis, which can be obviously repaired by individually driving the 4 coils. Note, however, that correction of parasitic aberrations is not possible by this strategy.

REVIEWERS' COMMENTS

Reviewer #1 (Remarks to the Author):

I think my concerns have been addressed at the best of Authors' possibility. I thin the paper should be published as it is.

Reviewer #2 (Remarks to the Author):

Dear authors, thank you for your answers to my comments, which I find satisfactory enough. I therefore recommend this article for publication in Nature communication.

Reviewer #3 (Remarks to the Author):

The authors addressed my questions. The work is ready for publication.